# Salivary Biomarkers and Oral Health in Liver Transplant Recipients, with an Emphasis on Diabetes

**DOI:** 10.3390/diagnostics11040662

**Published:** 2021-04-07

**Authors:** Annika Emilia Norrman, Taina Tervahartiala, Ella Sahlberg, Timo Sorsa, Hellevi Ruokonen, Lisa Grönroos, Jukka H. Meurman, Helena Isoniemi, Arno Nordin, Fredrik Åberg, Jaana Helenius-Hietala

**Affiliations:** 1Department of Oral and Maxillofacial Diseases, Helsinki University Hospital, University of Helsinki, P.O. Box 220, 00100 Helsinki, Finland; taina.tervahartiala@helsinki.fi (T.T.); timo.sorsa@helsinki.fi (T.S.); hellevi.ruokonen@hus.fi (H.R.); lisa.gronroos@hus.fi (L.G.); jukka.meurman@helsinki.fi (J.H.M.); jaana.s.helenius@helsinki.fi (J.H.-H.); 2Department of Oral and Maxillofacial Diseases, University of Helsinki, 00100 Helsinki, Finland; sahlberg.ella@gmail.com; 3Department of Dental Medicine, Karolinska Institutet, 141 04 Huddinge, Sweden; 4Department of Transplantation and Liver Surgery, Helsinki University Hospital, University of Helsinki, P.O. Box 372, 00100 Helsinki, Finland; helena.isoniemi@hus.fi (H.I.); arno.nordin@hus.fi (A.N.); fredrik.aberg@hus.fi (F.Å.)

**Keywords:** periodontitis, teeth, liver disease, oral infections, saliva, salivary biomarkers

## Abstract

Salivary biomarkers have been linked to various systemic diseases. We examined the association between salivary biomarkers, periodontal health, and microbial burden in liver transplant (LT) recipients with and without diabetes, after transplantation. We hypothesized that diabetic recipients would exhibit impaired parameters. This study included 84 adults who received an LT between 2000 and 2006 in Finland. Dental treatment preceded transplantation. The recipients were re-examined, on average, six years later. We evaluated a battery of salivary biomarkers, microbiota, and subjective oral symptoms. Periodontal health was assessed, and immunosuppressive treatments were recorded. Recipients with impaired periodontal health showed higher matrix metalloproteinase-8 (MMP-8) levels (*p* < 0.05) and MMP-8/tissue inhibitor of matrix metalloproteinase 1 (TIMP1) ratios (*p* < 0.001) than recipients with good periodontal health. Diabetes post-LT was associated with impaired periodontal health (*p* < 0.05). No difference between groups was found in the microbial counts. Salivary biomarker levels did not seem to be affected by diabetes. However, the advanced pro-inflammatory state induced by and associated with periodontal inflammation was reflected in the salivary biomarker levels, especially MMP-8 and the MMP-8/TIMP-1 molar ratio. Thus, these salivary biomarkers may be useful for monitoring the oral inflammatory state and the course of LT recipients.

## 1. Introduction

Post-transplant infections can lead to complications in liver transplant (LT) recipients. Therefore, it is essential that oral infections are diagnosed before admitting patients to the waiting list for an LT. Previous studies have demonstrated a significant association between post-LT systemic infections and a lack of pre-transplant dental treatment [1]. Furthermore, oral diseases notably influence chronic liver disease, which can, in turn, have an adverse effect on oral health [2,3]. Caries and periodontitis are the most common oral diseases among patients with chronic liver disease [4].

A systemic spread of dental infections seems to correlate with accelerated liver disease [5,6]. A previous study reported that periodontitis constitutes an independent risk factor for severe chronic liver disease [7]. LT recipients often have poor oral hygiene, and a high need for dental and periodontal treatment [8]. In general, the severity of liver disease is associated with poor oral health [9].

Liver disease and diabetes have an established association. Type II diabetes and insulin resistance are risk factors for the development of chronic liver disease, while diminished liver function due to cirrhosis can cause impaired glycemic control [10]. Post-transplant diabetes is a recognized complication of liver transplantation and associated with poor outcomes and increased mortality [11]. Furthermore, multiple studies show a bidirectional association between periodontitis and diabetes [12,13,14].

Saliva samples can be easily and non-invasively obtained, and contain numerous active biomolecules [15]. Therefore, saliva is increasingly being adopted in the diagnostics of oral and systemic diseases [16,17]. Salivary biomarkers have also been examined as a way to detect hepatocellular carcinoma [18]. Furthermore, salivary analysis has been suggested as a promising tool for assessing hepato-metabolic and cardiovascular complications in pediatric patients with obesity-related metabolic syndrome and fatty liver disease [19]. The use of saliva and oral fluids as a diagnostic tool in children following LT has been investigated as a potential non-invasive way to monitor these patients’ oral and systemic conditions [20]. Active matrix metalloproteinase-8 (aMMP-8), in particular, has been widely investigated, and is potentially up-regulated by diabetes [21,22,23,24]. Elevated aMMP-8 levels have also been associated with periodontitis [25]. As LT recipients are a difficult patient group due to their immunosuppressive state, saliva samples would be a simple and non-invasive method to monitor the health of this patient group. However, to the best of our knowledge, no studies have yet been performed regarding salivary biomarkers in adult LT recipients.

The aims of this retrospective study were to examine and compare periodontal health, oral microbial findings and salivary biomarkers in LT recipients with and without diabetes, and to compare the biomarkers and oral microbial findings in LT recipients with good and poor periodontal health. We hypothesized that LT recipients with diabetes would have poor periodontal health, and that this would have translated into higher levels of pro-inflammatory salivary biomarkers.

## 2. Materials and Methods

This study was approved by the Helsinki and Uusimaa Hospital District Ethics Committee (192/13/03/02/2008. 12 August 2008), and was performed in accordance with the Declaration of Helsinki. The LT recipients were informed about the study, and signed an informed consent form.

This study included 84 adults who received a LT between the years 2000 and 2006 at Helsinki University Hospital (HUS), which is the only national center for organ transplantation in Finland. Patients with a post-transplant oral examination recorded between the years 2008 and 2011 and a minimum follow-up of two years after LT were included in the study. The indications for transplantation were chronic liver disease (CLD), including alcohol liver disease, cholestatic liver disease, autoimmune hepatitis, viral hepatitis, and metabolic liver disease, or acute liver failure (ALF) [26].

### 2.1. Clinical, Radiographic, and Microbiological Oral Examination

Before LT, the recipients underwent clinical and radiological oral examinations. The recipients were re-called for a follow-up examination by experienced dental specialists (H.R. and L.G.) an average six years after transplantation (range 2–11 years). The examination involved recording the dental status, measuring periodontal pockets from six surfaces for each tooth, and recording oral mucosal lesions and gingival overgrowth. The oral mucosal findings of these 84 LT recipients were described in detail by Helenius-Hietala et al. [27]. A panoramic radiograph of the jaws was taken, and the recipients filled out a questionnaire about subjective oral symptoms, such as dry mouth, distortion of the sense of taste, difficulty swallowing, and symptoms of burning mouth. Smoking habits, alcohol use, oral hygiene practices, medications, and level of education were also recorded.

Periodontal health was evaluated using the periodontal inflammatory burden index (PIBI) [28], and the periodontal diagnosis was retrospectively reassessed according to the 2017 World Workshop on the Classification of Periodontal and Peri-Implant Diseases and Conditions [29]. Recipients were divided into the “low PIBI” (0 to 3; indicating good periodontal health) and “high PIBI” (>3; indicating impaired periodontal health) groups based on the median PIBI score of 3 (range 0 to 45). Six recipients lacked a periodontal diagnosis, due to being edentulous. Three recipients were missing PIBI values due to a lack of antibiotic prophylaxis at the time of examination. PIBI scores were calculated using dental data from the hospital records. The periodontal pockets measured were true periodontal pockets. In this assessment, the number of periodontal sites indicating moderate or advanced periodontitis was taken into account. Moderate periodontitis was defined as a periodontal pocket depth ≥ 4 mm and <6 mm at probing. Advanced periodontitis was defined as a periodontal pocket depth ≥ 6 mm. The final PIBI score was determined by adding the number of periodontal sites indicating moderate periodontitis to the weighed number of periodontal sites indicating advanced periodontitis [26,30].

The stimulated and unstimulated saliva secretion rates were measured [26]. The samples were collected between 9 a.m. and 2 p.m.; before collection the recipients refrained from eating and smoking in the morning. Unstimulated saliva was collected using the free flow method, and stimulated saliva samples were collected by having the patient chew a paraffin tablet for five minutes. Saliva samples were deep frozen for storage (−80 °C), and used later for biochemical analyses.

Subgingival pooled plaque samples were obtained from deep periodontal pockets using a sterile curette. The samples were cultivated and analyzed by routine methods in the HUS microbiological laboratory for the presence of Candida and the periodontal pathogens Prevotella intermedia, Micromonas micros, Tannerella forsythia, Porphyromonas gingivalis, and Parvimonas micra.

### 2.2. Analysis of Salivary Biomarkers

Stimulated saliva samples analyses were performed at the Biomedicum-1 research laboratory of the University of Helsinki. The analyses included immunoglobulins (Igs) A, G, and M, albumin, total amount of protein, and the cytokines tumor necrosis factor (TNF)-α and interleukin (IL)-1β, as well as matrix metalloproteinase (MMP)-8, tissue inhibitor of MMP (TIMP)-1, and the molar ratio between MMP-8 and TIMP-1. The analyses were performed as described by Lahdentausta et al. [31], Nylund et al. [30] and Rathnayake et al. [32].

The Igs, albumin, and cytokines were assayed using commercially available ELISA kits (Quantikine HS ELISA, R&D Systems, Minneapolis, MN, USA), according to the manufacturer’s instructions. The sample absorbance was measured using the VICTOR™ X3 Multilabel Plate Reader (Perkin Elmer, Waltham, MA, USA). TIMP-1 was analyzed using ELISA kits (Human Biotrak, ELISA System, GE Healthcare, Amersham Place, Buckinghamshire, UK) with a 1:10 sample dilution [31,32,33,34].

MMP-8 levels were measured using a time-resolved immunofluorometric assay (IFMA). Monoclonal MMP-8-specific antibodies 8708 and 8706 (Medix Biochemica, Espoo, Finland), a catching antibody and tracer antibody labelled with europium-chelate, were used. The assay buffer contained 20 mM Tris-HCl (pH 7.5), 0.5 M NaCl, 5 mM CaCl_2_, 50 μM ZnCl_2_, 0.5% bovine serum albumin, 0.05% sodium azide, and 20 mg/L diethylene-triamine-penta-acetic acid. The samples were diluted first in the assay buffer, and incubated for 1 h. Next, they were incubated with tracer antibody for one hour. Enhancement solution was added for five minutes, and the fluorescence was measured using an a1234 Delfia Research Fluorometer (Wallac, Turku, Finland) [30,32].

### 2.3. Patient Groups

The LT recipients were divided into three main groups for statistical analyses. For the first group, the LT recipients were divided into two sub-groups based on the diagnosis of diabetes post-LT. The recipients with diabetes had been diagnosed with either diabetes mellitus type I or II at the time of the oral examination. The time between the LT and the oral examination did not differ between recipients with and without diabetes post-LT. For the second group, the LT recipients were divided into two sub-groups based on their periodontal health, which in turn was based on their PIBI scores. For the third group, the LT recipients were first divided into six sub-groups based on their periodontal diagnosis: healthy periodontium, gingivitis, stage I and grade A/B/C periodontitis, stage II and grade A/B/C periodontitis, stage III and grade A/B/C periodontitis, and stage IV and grade A/B/C periodontitis. None of the recipients had stage IV periodontitis. The recipients were then grouped into three larger sub-groups based on the stage of inflammation and tissue destruction: healthy periodontium, gingivitis and stage I periodontitis, indicating inflammation and incipient tissue destruction, and stage II and III periodontitis indicating tissue destruction. Finally, the LT recipients were divided into sub-groups based on the use of immunosuppressive medication, including cyclosporin, tacrolimus, corticosteroids, mycophenolate mofetil (MMF), and azathioprine (AZA).

### 2.4. Statistical Analysis

The analyses were conducted using IBM SPSS Statistics 25 (IBM Corp., Armonk, NY, USA). None of the variables were normally distributed. Differences between groups were assessed using Pearson’s chi-squared test, the non-parametric Mann–Whitney U test, and the Kruskal–Wallis test, as appropriate. Univariate and multiple linear regression analyses were performed to study the factors associated with the PIBI score. Differences at a probability level of *p* < 0.05 were considered significant. Confidence intervals (CIs) were set at 95%.

## 3. Results

### 3.1. Basic Characteristics of LT Recipients with and without Diabetes

The basic characteristics of the 84 LT recipients are given in Table 1. Prior to the LT, 14 recipients (17% of the total) had a diabetes diagnosis (diagnosed 1 to 21 years pre-LT, mean 9 years, SD ± 8). Recipients with diabetes took significantly more daily medications and had a lower unstimulated salivary flow rate than those without diabetes. A significantly higher number of LT recipients who did not develop diabetes used corticosteroids compared to those with diabetes. Seven recipients (8% of the total) developed diabetes after transplantation. Of the 21 recipients with diabetes post-LT, four had diabetes mellitus type I and 17 had diabetes mellitus type II. Interestingly, there was no statistically significant difference between the groups regarding their self-assessed oral health.

### 3.2. Periodontal Health and Salivary Biomarkers in LT Recipients with and without Diabetes

The periodontal status and presence of periodontal pathogens in LT recipients with or without diabetes are given in Table 2. The number of periodontal pockets ≥ 4 mm was significantly higher in LT recipients with diabetes. However, we found no significant differences between the groups regarding the other oral health parameters, or in the number of periodontal pathogens between the groups. When comparing salivary biomarkers between LT recipients with and without diabetes, we found no significant differences between the groups. The results are given in Appendix A.

### 3.3. Salivary Biomarkers and Salivary Flow Rates in LT Recipients with Good and Impaired Periodontal Health

Table 3 compares the salivary biomarkers and salivary flow rates between LT recipients with low vs. high PIBI scores, and with different stages of periodontal disease. Those with high PIBI scores had significantly higher mean MMP-8 levels, higher MMP-8/TIMP-1 molar ratios, and higher stimulated salivary flow rates than recipients with a lower PIBI score. The TIMP-1 levels were significantly higher in the gingivitis and stage I group than in the stage II and III group. The healthy periodontium group had significantly higher IgM levels compared to the stage II and III group. Furthermore, the stimulated salivary flow rate was significantly lower in the healthy periodontium group than in the stage II and III group. Interestingly, the mean level of MMP-8 was highest in the healthy periodontium group and lowest in the stage II and III group, although these differences were non-significant.

In the univariate linear regression analysis, the MMP-8/TIMP-1 molar ratio (*p* = 0.001) and stimulated salivary flow rate (*p* = 0.017) were the only significant predictors of a high PIBI score. Both the MMP-8/TIMP-1 molar ratio (*p* < 0.001) and stimulated salivary flow rate (*p* = 0.002) remained significant in the multiple regression analysis. Results of the regression analyses are given in Appendix A.

The presence of periodontal pathogens and Candida and/or other yeasts in LT recipients with a low PIBI score vs. high PIBI score was also studied. We found no significant differences between the groups in the detection rates for any of the microorganisms analyzed (data not shown).

### 3.4. Periodontal Health in Patients with and without Diabetes

In analyzing the association between diabetes and PIBI scores, we found that the PIBI score of LT recipients with diabetes was significantly higher than the PIBI score of LT recipients without diabetes. In addition, a significantly higher percentage of LT recipients with diabetes had a high PIBI score. Nevertheless, when analyzing the distribution of the stage of periodontitis between these two groups, we found no significant difference. However, it should be noted that none of the recipients with diabetes had a healthy periodontium, while 17% of the recipients without diabetes had a healthy periodontium. The results are shown in Table 4.

### 3.5. Salivary Biomarkers and Periodontal Health Stratified According to Immunosuppressive Medication

When comparing salivary biomarkers and PIBI scores between LT recipients using different immunosuppressive medications, we found that the total concentration of proteins in the saliva of recipients using tacrolimus was significantly higher than that for the recipients using cyclosporin A (*p* = 0.003). The salivary MMP-8 (*p* = 0.013) and TNF-α (*p* = 0.042) levels were significantly higher in recipients not taking corticosteroid medication compared to those on corticosteroids. When comparing the groups of MMF, AZA, and no MMF or AZA, a significant difference could be seen in the amount of IL-1β (*p* = 0.015) and TNF-α (*p* < 0.001). Recipients taking AZA had the highest levels of both biomarkers. Furthermore, recipients taking MMF and AZA medications had a higher PIBI score compared to recipients on neither of these medications (*p* = 0.045). The results are given in detail in Appendix A.

## 4. Discussion

Contrary to our hypothesis, diabetes in LT recipients did not seem to affect oral inflammatory marker levels or microbial findings. The marker levels mainly reflected impaired periodontal health. Thus, the main finding of this study was that higher MMP-8 levels and a higher MMP-8/TIMP-1 molar ratio correlated significantly with higher PIBI scores. A higher PIBI score correlated with diabetes. 

The life-long immunosuppressive medication taken by LT recipients exposes them, for example, to cardiovascular complications, metabolic syndrome, and diabetes [35,36]. Only 8% of the LT recipients in our study developed diabetes post-LT. Diabetes has been shown to act as a bidirectional risk factor for periodontitis and oral diseases due to the prolonged inflammatory state induced by up-regulated inflammatory mediators such as MMP-8, leading to increased periodontal tissue destruction [21,22,23]. Our results showed that when using PIBI scoring to assess periodontal health, LT recipients with diabetes and high PIBI scores were found to have impaired periodontal health. However, when using the periodontal staging and grading system, no significant difference between LT recipients with or without diabetes was observed. This might, however, be explained by the small group size. Nevertheless, none of the LT recipients with diabetes had a healthy periodontium and a greater percentage had a higher stage of periodontitis compared to LT recipients without diabetes. Hyperglycemia in patients with diabetes not only predisposes them to periodontal disease and related low-grade systemic inflammation, but also to other oral complications, such as dental caries, candidiasis, burning mouth syndrome, salivary dysfunction, and xerostomia [37]. Periodontal treatment is beneficial for both local and systemic health, especially with diabetic patients [38,39]. Together with diabetes, the organ transplantation and the underlying liver disease complicates oral health; this emphasizes the importance of periodontal treatment in these patients.

High oral fluid MMP-8 levels have been linked to periodontal disease [40,41]. As MMP-8 has been suggested for use as diagnostic aid for periodontitis, it may also be useful in the diagnosis of systemic conditions, such as diabetes [21,22,23,24]. We found the MMP-8 levels in LT recipients with diabetes to be higher than in recipients with no diabetes, although the difference was statistically non-significant. In addition to MMP-8, patients with severe periodontitis have been reported to have significantly higher levels of IL-1β and an elevated MMP-8/TIMP-1 molar ratio [32]. TIMPs, as the inhibitors of MMPs, control the local activity of MMPs in tissues [42]. On the one hand, our results align well with these findings, as we found a higher level of MMP-8 and a higher MMP-8/TIMP-1 molar ratio in recipients with a higher PIBI score and, therefore, impaired periodontal health. On the other hand, there were no significant differences in these levels when comparing the stages of periodontal disease. Significantly higher levels of TIMP-1 were observed in LT recipients with a healthy periodontium, compared to recipients with stage II or III periodontitis, in our study. Interestingly, LT recipients with more advanced periodontitis had the lowest levels of MMP-8, although the differences were non-significant. All infection foci, including periodontitis, were treated before the patients were admitted for liver transplantation to our hospital. This could explain the low levels of MMP-8 among the LT recipients with advanced periodontitis, as periodontal treatment is known to lower MMP-8 levels [41,43]. Moreover, our results showed significantly higher salivary levels of IgM in recipients with better periodontal health. Patients with primary biliary cholangitis have higher levels of plasma and serum IgM levels, possibly explaining these differences [44].

When investigating the relationship between salivary biomarkers and immunosuppressive medications, we found that LT recipients who were taking corticosteroid medication had significantly less salivary MMP-8 and TNF-α. As MMP-8 and TNF- α are pro-inflammatory mediators, this result could be explained by the strong anti-inflammatory effect of corticosteroids. The amounts of TNF-α and IL-1β differed significantly between the recipients using and not using AZA or MMF. Recipients using AZA had the highest, and those using MMF the lowest, levels of these biomarkers. However, because AZA is most commonly used by patients whose underlying cause for LT is autoimmune liver disease, these results could in part depend on the indication for the medication, and not merely on the use of the medication. Unfortunately, our sample size was too small to study this matter in more detail. Recipients using AZA or MMF had, nevertheless, higher PIBI scores, indicating a worse periodontal health compared with the recipients who did not take these drugs. Previous studies have indicated an association between periodontal parameters and immunosuppressive medications, a difference in the prevalence of several potentially periodontal pathogenic bacteria, and a shift in the subgingival biofilm in patients taking an immunosuppressive medication [45]. This, however, could not be verified in the present study. Furthermore, the use of mechanistic target of rapamycin inhibitors is rare in Finland, and thus their role could not be investigated in this study.

Finally, we decided to include two parameters for determining periodontal health in this study. As the 2017 World Workshop on the Classification of Periodontal and Peri-Implant Diseases and Conditions [29] is the preferred standard, we included this classification in our study. However, due to the small sample size being a limitation, we also included the PIBI scoring as a parameter for periodontal health, allowing us to use larger sub-groups.

The strength of this study lies in it being the first investigation where comprehensive data on oral health, salivary biomarkers, microbiological findings, and the results from a self-assessment of an oral health questionnaire could be analyzed. Despite including all eligible patients to the study from the only transplant hospital in Finland who had been operated on between the years 2000 and 2006, the number of patients included remained small. The lack of a healthy control group is a major limitation, but due to practical reasons controls could not be included. Moreover, samples for salivary biomarker analyses had not been taken before the LT. Furthermore, we were unable to include blood glucose levels in our data analysis. However, we believe that our results are important for further hypothesis generation, for a larger and preferably multicentric investigation.

## 5. Conclusions

Our results show that LT recipients with high levels of pro-inflammatory salivary biomarkers, such as MMP-8 or the MMP-8/TIMP-1 molar ratio, tend to have a higher periodontal inflammatory burden index. Furthermore, LT recipients with diabetes had poorer periodontal health. However, diabetes in the LT recipients did not seem to affect the salivary biomarker levels. Thus, an inflammatory state induced by periodontal inflammation is seen in these salivary biomarker levels. As saliva is non-invasively and easily collected, assessing the oral inflammatory state from saliva samples could be a useful diagnostic tool and a good addition to conventional methods used for clinical decision-making in this difficult patient group.

## Figures and Tables

**Table 1 diagnostics-11-00662-t001:** Background, medications, and oral symptoms of liver transplant recipients with and without diabetes, after liver transplantation.

Parameter	No Diabetes	Diabetes	*p*
Total *n* of patients	63	21	
Age at oral exam post-LT [years] ^1^	53.4 (24.6–70.9)	59.4 (42.9–69.4)	0.054
Women/men [%]	44/56	24/76	0.124
CLD/ALF [%]	71/29	91/10	0.136
Diabetes type [%]			
DM type I		19	
DM type II		81	
Smoking [%]	19	5	0.169
Alcohol use [%]	32	38	0.790
Educational level [%]			
University	18	10	0.497
Technical school	27	38	0.406
Other	55	52	1.000
Working status [%]			
Working full-time	40	38	1.000
Unemployed	7	0	0.352
Retired	53	62	0.613
Cardiovascular disease [%]	49	71	0.076
Number of medications ^2^	6.3 (2.3)	8.1 (1.7)	0.002
Medications [%]			
Cardiovascular	87	95	0.439
Pulmonary	6	5	1.000
CNS	21	10	0.336
Analgesic	16	10	0.721
Immunosuppression [%]			
Cyclosporine	46	38	0.211
Tacrolimus	38	62	0.077
mTOR inhibitor	6	0	0.568
Corticosteroid	32	10	0.048
Azathioprine	13	19	0.721
Mycophenolate mofetil	35	33	1.000
Xerostomia [%]	48	43	0.803
Dysphagia [%]	21	14	0.750
Burning mouth syndrome [%]	18	10	0.502
Dysgeusia [%]	3	5	1.000
Unstimulated salivary flow rate [mL/min] ^2^	0.5 (0.4)	0.2 (0.2)	0.031
Stimulated salivary flow rate [mL/min] ^2^	1.7 (1.0)	1.8 (1.1)	0.948
Self-assessment of oral health, good [%]	38	48	0.450

Abbreviations: ALF = acute liver failure; CLD = chronic liver disease; CNS = central nervous system; DM = diabetes mellitus; LT = liver transplantation; mTOR = mechanistic target of rapamycin. *p*-values correspond to Pearson’s chi-squared test or the Mann–Whitney U test as appropriate. ^1^ The data are presented as the median (range). ^2^ The data are presented as the mean (SD).

**Table 2 diagnostics-11-00662-t002:** Periodontal status and presence of periodontal pathogens in liver transplant recipients with or without diabetes.

Periodontal Status	No. of LT Recipients	No Diabetes	Diabetes	*p*
No. of LT recipients	84	63	21	
No. of teeth post-LT ^1^	81	21.7 (8.9)	21.7 (7.1)	0.431
No. of teeth extracted pre-LT ^1^	69	2.8 (8.9)	2.8 (3.8)	0.638
No. of periodontal pockets ^1^	75			
4–5 mm periodontal pockets		5.2 (6.9)	8.5 (5.6)	0.007
≥6 mm periodontal pockets		0.4 (1.2)	0.4 (1.2)	0.711
No. of furcation lesions ^1^	75	0.3 (0.7)	0.7 (1.7)	0.333
No. of vertical bony pockets ^1,3^	66	0.5 (1.2)	0.1 (0.2)	0.098
Alveolar bone loss ^2,3^	84			
None		24 (38.1)	5 (6.0)	0.295
Cervical third of root		29 (46.0)	13 (61.9)	0.314
Mid third of root		8 (12.7)	3 (14.3)	1.000
Apical third of root		2 (3.2)	0 (0)	1.000
Periodontal pathogens ^2^	78			0.796
Yes		28 (48.3)	11 (55.0)	
No		30 (51.7)	9 (45.0)	

Abbreviations: LT = liver transplantation. ^1^ Results are given as mean (SD), *p*-values correspond to Mann–Whitney U test. ^2^ Results are given as *n* (%) and *p*-values correspond to Pearson’s chi-squared test ^3^ Alveolar bone loss and vertical bony pockets were calculated from panoramic tomography x-rays of the jaws.

**Table 3 diagnostics-11-00662-t003:** Salivary biomarkers and salivary flow rates in liver transplant recipients with good and impaired periodontal health and with and without periodontal disease.

Biomarker/Parameter	Low PIBI	High PIBI	*p*	Healthy Periodontium	Gingivitis & Stage I Periodontitis	Stage II & III Periodontitis	*p*
No. of patients	43	38		10	43	25	
MMP-8 [ng/mL]	144.6 (151.0)	173.9 (89.3)	0.023	221.0 (118.5)	163.2 (131.4)	140.2 (101.1)	0.219
TIMP-1 [ng/mL]	282.5 (225.7)	212.3 (145.4)	0.056	335.3 (395.3)	268.0 (151.8)	170.5 (105.9)	0.003 ^1^
MMP-8/TIMP-1 molar ratio	0.2 (0.2)	0.5 (0.3)	<0.001	0.4 (0.2)	0.3 (0.3)	0.4 (0.3)	0.128
Total protein [mg/mL]	1.5 (0.4)	1.4 (0.5)	0.638)	1.4 (0.4)	1.4 (0.4)	1.4 (0.5)	0.890
Albumin [μg/mL]	52.0 (66.8)	48.7 (39.3)	0.624	68.9 (58.1)	59.2 (66.5)	38.0 (30.2)	0.119
IgA [μg/mL]	43.6 (51.9)	42.6 (35.3)	0.382	59.0 (82.5)	44.3 (42.0)	38.9 (22.5)	0.876
IgG [μg/mL]	19.4 (20.7)	18.1 (15.1)	0.918	24.7 (20.7)	13.5 (18.9)	16.9 (16.6)	0.247
IgM [μg/mL]	6.6 (15.7)	2.9 (3.1)	0.992	5.7 (5.4)	5.1 (12.0)	1.9 (1.5)	0.012 ^2^
IL-1β [pg/mL]	203.3 (206.0)	233.0 (150.5)	0.112	279.5 (165.0)	228.9 (207.0)	222.9 (147.1)	0.484
TNF-α [pg/mL]	7.2. (16.6)	6.6 (9.2)	0.753	2.5 (2.9)	7.8 (16.0)	7.6 (11.4)	0.819
Unstimulated salivary flow rate [mL/min]	0.4 (0.4)	0.4 (0.3)	0.619	0.5 (0.7)	0.4 (0.3)	0.4 (0.4)	0.775
Stimulated salivary flow rate [mL/min]	1.5 (1.0)	2.0 (1.0)	0.045	1.2 (0.7)	1.6 (0.9)	2.3 (1.2)	0.014 ^3^

Abbreviations: PIBI = periodontal inflammatory burden index. *p*-values for PIBI groups correspond to the Mann–Whitney U test or Kruskal–Wallis test, as appropriate. Liver transplant recipients with a low PIBI score (0–3) were considered to have a good periodontal health, and recipients with a high PIBI score (4–45) were considered to have impaired periodontal health. ^1^ Stage II and III group significantly differs from the Gingivitis + Stage I group ^2^ Stage II and III group significantly differs from the Healthy Periodontium group ^3^ Healthy Periodontium group significantly differs from the Stage II and III group.

**Table 4 diagnostics-11-00662-t004:** Distribution of stage of periodontitis and periodontal inflammatory burden index in liver transplant recipients with no diabetes and with diabetes, after liver transplantation.

Periodontal Stage	No Diabetes	Diabetes	*p*
No. of LT recipients	63	21	
Healthy periodontium ^1^	10 (17.2)	0 (0)	
Gingivitis ^1^	16 (27.6)	5 (25.0)	
Stage I, Grade A/B/C periodontitis ^1^	18 (31.0)	4 (25.0)	
Stage II, Grade A/B/C periodontitis ^1^	6 (10.3)	6 (30.0)	
Stage III, Grade A/B/C periodontitis ^1^	8 (13.8)	5 (25.0)	
Stage IV, Grade A/B/C periodontitis ^1^	0 (0)	0 (0)	
PIBI score ^2^	5.5 (8.2)	8.8 (6.9)	0.011
Low PIBI score (0–3) ^1,3^	38 (62.3)	5 (25.0)	0.005
High PIBI score (4–45) ^1,3^	23 (37)	15 (75.0)	

Abbreviations: PIBI = periodontal inflammatory burden index. ^1^ Results are given as *n* (%) of patients in each group. ^2^ Results are given as the mean (SD). ^3^
*p*-values correspond to Pearson’s chi-squared test. Pearson’s chi-squared test showed no significant difference (*p* = 0.058) in the distribution of the stage of periodontitis between the patients with diabetes and without diabetes.

## Data Availability

The data presented in this study are available on request from the corresponding author. The data are not publicly available due to their confidential nature.

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
