# Peer review of "Salivary Biomarkers and Oral Health in Liver Transplant Recipients, with an Emphasis on Diabetes"

_diagnostics, 2021, doi:10.3390/diagnostics11040662_

Round 1

Reviewer 1 Report

The research doesn't seem logical. If the authors aim to compare the parameters of saliva in patients with and without diabetes, then I did not see these data in the article. The assessment of the level of oral hygiene and the severity of periodontal diseases is given and the characteristics of saliva are given, however, for the case with diabetes / no diabetes, no data are available. In general, inflammatory periodontal diseases are a consequence of diabetes mellitus and, apparently, should correlate with glucose levels, but I also do not see such data in the article. If we are talking about patients after liver transplantation, then the obtained data should be compared with the corresponding groups (there is diabetes / no diabetes, a different state of the oral cavity), who have not received a liver transplant. 

Reviewer 2 Report

The authors summarized the current knowledge about the remote effects of oral infections and the relationship between periodontal status and impairing effect of poor liver function. The clinical study or M&M are well-structured and documented.  The conclusion is adequate, however, in the clinical practice it has a low significant since the investigation of the level of these biomarkers is not part of routine dental examination and not available everywhere.

I have a question regarding the immunosuppressant agents. Mammalian target of rapamycin inhibitors may play a role in the development of osteonecrosis of the jaw. Was it investigated in this clinical study, or did you notice this severe complication in patients treated with mTOR inhibitors?

With this completion the manuscript is acceptable for publication.

Round 2

Reviewer 1 Report

The authors provided detailed answers to the reviewer's questions. However, in essence, the answers are reduced to the rationale why the relevant data is missing in the article. So, I think it is necessary that there is a comparison of the studied parameters in the groups of diabetics / non-diabetics without liver transplantation. This is necessary to show what changes are caused by transplantation. The authors called the lack of these data a limitation of the study. If the editor sees fit, then you can publish with these restrictions.

Author Response

Helsinki, April 3rd, 2021

Manuscript ID: diagnostics-1162683

Title: Salivary biomarkers and oral health in liver transplant recipients, with emphasis on diabetes

Point-by-point response to Reviewer 1 Comments, round 2

Point 1: The authors provided detailed answers to the reviewer's questions. However, in essence, the answers are reduced to the rationale why the relevant data is missing in the article. So, I think it is necessary that there is a comparison of the studied parameters in the groups of diabetics / non-diabetics without liver transplantation. This is necessary to show what changes are caused by transplantation. The authors called the lack of these data a limitation of the study. If the editor sees fit, then you can publish with these restrictions.

Response 1: We thank the reviewer for the comment. We agree, with the reviewer that a control group without a liver transplant could further strengthen the findings in our study. However, we do think our current study design is adequate. Defining and recruiting a control group is not a simple task, and it would further complicate the study design. For instance, examining a control group without liver transplants is not necessarily sufficient alone, as the potential degrees of liver insufficiency and immunosuppression within the group would have to be adjusted for. Furthermore, as the main aim of our study is to examine and compare periodontal health, oral microbial findings and salivary biomarkers in liver transplant recipients with and without diabetes, adding a control group consisting of patients without a transplant would not necessarily make the study address the aim better.

Considering these aspects, we thus respectfully disagree with the reviewer, and do not think an additional control group of patients without liver transplants is strictly necessary, as we already compare liver transplant recipients with and without diabetes and good and impaired periodontal health in this study.